# Beads for Cell Immobilization: Comparison of Alternative Additive Manufacturing Techniques

**DOI:** 10.3390/bioengineering10020150

**Published:** 2023-01-23

**Authors:** Maria Laura Gatto, Paolo Mengucci, Daniel Munteanu, Roberto Nasini, Emanuele Tognoli, Lucia Denti, Andrea Gatto

**Affiliations:** 1Department DIISM, Università Politecnica delle Marche, Via Brecce Bianche 12, 60131 Ancona, Italy; 2Department SIMAU, Università Politecnica delle Marche, Via Brecce Bianche 12, 60131 Ancona, Italy; 3Material Science Department, Transilvania University of Brasov, 29 Eroilor Blvd., 500036 Brasov, Romania; 4Prosilas S.r.l., Via Terracini 14, 60212 Civitanova Marche (MC), Italy; 5Department of Engineering “Enzo Ferrari”, Università di Modena e Reggio Emilia, Via P. Vivarelli 10, 41125 Modena, Italy

**Keywords:** polymer beads, additive manufacturing, industrial scale

## Abstract

The attachment or entrapment of microbial cells and enzymes are promising solutions for various industrial applications. When the traps are beads, they are dispersed in a fluidized bed in a vessel where a pump guarantees fresh liquid inflow and waste outflow without washing out the cells. Scientific papers report numerous types of cell entrapment, but most of their applications remain at the laboratory level. In the present research, rigid polymer beads were manufactured by two different additive manufacturing (AM) techniques in order to verify the economy, reusability, and stability of the traps, with a view toward a straightforward industrial application. The proposed solutions allowed for overcoming some of the drawbacks of traditional manufacturing solutions, such as the limited mechanical stability of gel traps, and they guaranteed the possibility of producing parts of constant quality with purposely designed exchange surfaces, which are unfeasible when using conventional processes. AM proved to be a viable manufacturing solution for beads with complex shapes of two different size ranges. A deep insight into the production and characteristics of beads manufactured by AM is provided. The paper provides biotechnologists with a manufacturing perspective, and the results can be directly applied to transit from the laboratory to the industrial scale.

## 1. Introduction

The immobilization of microbial cells and enzymes is an increasingly applied solution in biotechnological processes. It has led to considerable research for industrial applications, such as food industries, enzyme biosynthesis [1], hydrocarbon decontamination [2], and wastewater biotreatment [3]. Immobilization is a general term that describes many different forms of cellular attachment or entrapment, including flocculation, surface adsorption, covalent bonding with load-bearing media, cell crosslinking, encapsulation in a polymer gel, and entrapment in a matrix [1]. In the case of a reactor using beads for cell immobilization, physical media are dispersed in a fluidized bed. A pump guarantees the vessel inflow of fresh liquid and the outflow of waste by remaining below the cell wash-out flow. For industrial applications, the cell immobilization units are required to be inexpensive, stable, reusable, inert, and biocompatible. The stability of the beads is vital for maintaining a high substrate-to-product conversion [4]. In contrast, a high cell density in a bioreactor guarantees productivity [5], with minimal internal mass transfer limitations. Calcium alginate is the most common immobilization matrices used today (e.g., for wastewater treatment) [6,7], followed by polyvinyl alcohol (PVA) [8,9]. Entrapment is the most-used immobilization technique. Regarding the beads, Lopez et al. [1] showed the pronounced effect on the stability of the process using the diameter of the bead, where they highlighted that beads of 1 mm in diameter showed the maximum transformation efficiency. There are several immobilization solutions, and Verbelen et al. [10] classified four immobilization categories:attachment to a surfaceentrapment within a porous matrixcontainment behind a barrierself-aggregation

Even though the literature reports on numerous types of matrices and beads being locked up in a fluidized bed, as early as 2004, Junter and Jouenne [11] reported on how a vast majority of the application studies remained at the laboratory scale [12]. Polymeric beads (e.g., beads made of polyurethane, polystyrene, and polyvinyl alcohol) are an alternative to a porous matrix. Furthermore, using rigid polymers may be beneficial for overcoming some of the limitations of traditional manufacturing solutions, such as the poor mechanical stability of gel traps [13]. Beads are typically spherical, with a diameter range of 0.3–5 mm, but there is no accordance regarding suitable bead dimensions. Douglas et al. [14] suggested using spherical beads with a diameter of <1 mm for an animal cell culture. Conventional manufacturing solutions are based on the breakup of a liquid capillary jet or extrusion by various cutting techniques (vibration, mechanical action, or air flow), inducing the drops to fall into a liquid bath. For example, conventional manufacturing techniques allow for the production of 0.2–5 mm-diameter alginate beads [14], but they are unsuitable for producing parts of consistent quality with designed exchange surfaces, and their production is expensive [10].

The complex shapes, dimensions, and mechanical resistance required by the beads make AM a viable manufacturing solution for their production. Despite this potential, only a few papers have investigated the bead manufacturing process and the accuracy/deviation versus the nominal shape of such beads. A majority of the literature deals with the efficiency and biological mechanisms involved in using the beads. Dong [15] and Belgrano [2] produced polyamide beads with diameters of 15 and 30 mm, respectively, using powder bed fusion. They proved that the nutrient adsorption efficiency of the cells depends on features such as the size, porosity, and surface characteristics of the traps. Other authors have investigated AM capacity using a bioprinter for manufacturing hydrogel parts [16], but the instability of the hydrogel process and its difficulty in controlling viscosity make this solution unsuitable for upgrading it to the industrial scale. However, in order to transition from the laboratory scale to industrial applications, it is necessary to overcome the lack of knowledge regarding the dimensional performance, tolerance limits, and surface characteristics of AM bead production [17]. The minimum resolution and minimum layer thickness of AM machines in the literature are often adopted as discriminating capability factors for producing small devices [18]. These parameters are suitable for describing the system performance, but they are inadequate for foretelling the accuracy of a three-dimensional object. This paper aimed to evaluate the manufacturing capacity of AM devices for bead production, identifying the manufacturing process limits and helping biotechnologists to choose a suitable manufacturing solution. The experimental plan included two different lines: (a) 3.0, 2.0, and 1.5 mm-diameter beads manufactured by vat photopolymerization using acrylic acid ester resin, each with and without a 10% and a 20% hydroxyapatite (HA) charge; (b) 15 and 30 mm-diameter polyamide (PA) beads manufactured using powder bed fusion (PBF). The latter builds on a promising literature study [2] by offering a deeper insight into the issues related to de-powdering and the accuracy of the manufactured parts versus the nominal geometry. Since the dimensions achievable by PBF are generally above the optimal range for immobilization beads [14], vat photopolymerization is also studied as an alternative solution that accomplishes smaller details and better accuracy. Regarding the biocompatibility of the two material systems, PA is widely recognized as being safe for use and for coming into contact with various human cells and tissues [19,20], though the approval for the photopolymerized acrylic acid ester that is being proposed for the production of dental prostheses is less mature [21].

## 2. Materials and Methods

The experimental plan included two different lines:

3.0, 2.0, and 1.5 mm-diameter beads manufactured by vat photopolymerization (DWS 029X, DWS, Thiene, Italy) using unreinforced and charged (10 wt.% and 20 wt.% hydroxyapatite) acrylic acid ester resin (Vitra DL375)15 and 30 mm-diameter lattice beads manufactured by PBF (EOS Formiga P110, EOS, Krailling, Germany) using polyamide 12, with three different repetitive units and three strut dimensions

Two material/process combinations were thus considered, both of which were within the optimal size range. The two AM systems are sketched in Figure 1.

Table 1 shows the nominal mechanical performance measures of both materials [24,25]. The range of the mechanical characteristics indicates the influence of the AM process parameters, whose optimization is the subject of extensive debate. Additives and reinforcements lead to variations in mechanical behavior, and they may cause the mechanical strength to decrease [26].

### 2.1. Beads Manufactured by Vat Photopolymerization

The fragile and sensible-to-tensile stress of acrylic acid ester has a 1% polymerization shrinkage that causes residual stresses; therefore, it requires a specific manufacturing strategy for avoiding deformations and ruptures. The manufacturing strategies consisted of arranging the beads in continuous chains of balls either built along a spiral (Figure 2) or supported by a linear trabecular structure (Figure 3). The spiral and linear layouts allow for an even distribution of the removable supports without dispersing single beads, and they guarantee the repeatability and handling of the beads. Both solutions minimize the support contact area, but the trabecular support structure is optimal in its ease of removal while the continuous spiral chain minimizes production time. The preliminary tests demonstrated that the support type has no measurable effects on the dimensional characteristics of the produced parts. The vat photopolymerization method involved the following post-processing steps:washing in 96% ethyl alcohol with the aid of compressed air to remove the unpolymerized resin from the as-built partfurther ultrasonic washing in ethyl alcohol to ensure the unpolymerized resin removal from the bead cavityUV oven post-curing for 20 minutesremoval of the supports followed by a surface finish

Table 2 and Table 3 show the dimensions of the spherical beads and the process parameters used. The geometry of the beads to be tested was chosen by merging the recognized requirements in terms of bead diameter, hole size, and exchange surface [10,13,14] with the resolution capability of the machine. The parameters in Table 3 were set based on industrial know-how. The beads are hollow spheres with eight holes that connect the inner cavity to the outer region. An important parameter for the exchange performance is the ratio between the exchange surface (i.e., the sum of the area of the eight holes) and the containment surface, which consists of the surface extension of the inner cavity reduced by the area of the holes (Equation (1)).
(1)$$exchange\;ratio=\frac{exchange\;surface}{containment\;surface}$$

Preliminary tests were performed to verify the feasibility of using 200 µm holes. The results showed that this diameter was at the lower boundary of the machine’s capability. Some of the holes were thorough and well-defined, while occasionally, the holes were closed. Therefore, in the design of beads, the hole dimensions were increased to 500 µm for the 3 and 2 mm-diameter beads and to 350 µm for the 1.5 mm-diameter beads.

In the simplified relationship reported in Equation 2 [27], E_c_ indicates the energy required to overcome the critical threshold of the polymerization activation process (mJ/mm^2^). The penetration coefficient D_p_ is an experimentally determined characteristic of the resin, and it expresses the distance at which the irradiance decreases by 1/e^2^ (37%). Therefore, the energy E required to reach the critical value (and thus, to trigger the polymerization) at a depth z_p_ is provided by:(2)$$E=E_{c}*e^{\left(\frac{z_{p}}{D_{p}}\right)}$$

### 2.2. Beads Manufactured by PBF

Table 4 shows the shape and dimensions of beads produced by PBF, while Table 5 provides the process parameters used. The chosen dimensions made the models comparable with those used by Belgramo [2] to investigate the bead performance for glucose transformation.

### 2.3. Characterization

Eleven different types of beads were produced using PBF while ten types of beads were produced by vat photopolymerization.

The diameter of the beads was measured by a Nikon SMZ1270 (Nikon, Tokyo, Japan) optical microscope. With respect to previous studies on alginate beads of simple shapes [1,14], in this research, optical microscopy was applied for a full dimensional characterization of the beads’ geometry, similar to the procedure used by Cui et al. [28]. The bead diameter was obtained using 10 to 15 points on the circumference, with three measurement repetitions. The outer surface roughness was measured on three different areas (853 × 511 mm^2^) for each specimen using the confocal head on a Nikon Eclipse LV150N (Nikon, Tokyo, Japan) optical microscope and using a 200× magnification objective with a Z scanning interval of 0.20 mm. The point data were post-processed by Mountains Map software using a Gauss filter set of 250 mm. The surface roughness has been recognized to have an effect on the first steps of the interaction between the cells and the entrapment media [15].

The hole replicas were obtained using the STC SHAPE-IN injection method. This method consists of pouring bicomponent silicone material (poly-vinyl siloxane) at room temperature, and after curing for approximately 50 min at room temperature, a silicone rubber replica (which has high elasticity) is obtained. It is important to ensure that the silicone material completely fills the bead at the holes, and for this purpose, the injection is continued until the silicone material fills half of the bead. As a consequence, to allow the extraction of the replica, the bead is broken. The optical measurement procedure was also applied to the replicas in order to verify the hole accuracy. This approach allowed for a dimensional verification of the hole along its entire length as direct measurements of the beads would only provide information about their outer surfaces.

Both bead typologies produced by PBF and vat photopolymerization were investigated using scanning electron microscopy (SEM) [28,29] and X-ray micro-computed tomography (XmCT). The surface morphology of the vat photopolymerized beads in charged and uncharged conditions was observed by SEM with a Tescan Vega 3 (Tescan Company, Brno, Czech Republic). The beads’ morphometric parameters were experimentally obtained by XmCT analysis carried out by a Bruker Sky scan 1174 (Bruker, Kontich, Belgium) system. Shadow images were acquired at V = 40 kV and I = 800 µA with a Cu-Kα radiation source, a pixel size of 16.4 µm, a rotation step of 0.3°, a total rotation angle of 180°, and an exposure time per projection of 1.8 s. The projections were processed by the NRecon reconstruction program (Bruker), collecting a stack of cross-sectional slices generated with the following settings: smoothing = 2, ring artefacts reduction = 3, and beam hardening correction = 10%. The morphometric parameters quantified for the PBF beads with a K-shape using CTAnalyzer software (Bruker, Kontich, Belgium) were: average strut thickness (mm), average pore size (mm), and closed and open porosity (%). Closed porosity describes the micro-porosity inside the struts, and it does not contribute to a bead’s open porosity; rather, it depends on the process parameters and influences the material density. On the other hand, open porosity refers to macro-porosity and may depend on the design of the beads [30]. Morphometric analysis of the K-shape beads was carried out for each sample on 3 cubic volumes of interest (VOI) of an 8 mm-side, which was selected in a residual powder-free region. Moreover, the presence of residual powder was confirmed in the PBF beads with K-shapes by cross-sectional slice analysis. The volume of the residual powder was quantified and 3D-reconstructed using Dragonfly software (Version 2020.1; Object Research Systems, Montreal, QC, Canada). 3D models of the manufactured beads were created using CTVox software (Bruker, Kontich, Belgium).

## 3. Results and Discussion

### 3.1. Beads Manufactured by Vat Photopolymerization

Figure 4 shows the diameters of the beads measured by optical microscopy. The accuracy of the beads produced with 10% HA-charged-resin was greater than those produced with the unreinforced material, and this can also be explained by the assumption that the HA behaved as a UV blocker. Hydroxyapatite at a low concentration can induce an effect similar to that of a laser with a smaller spot or of a decrease in the molar extinction coefficient. A further increase in the HA charge percentage caused a decrease in the efficiency of the process and in the possibility for controlling its parameters.

The measurements of the hole diameters showed a different shape from the nominal circular one. In fact, the hole had a more elliptical shape, with values ranging from a minimum of 0.43 ± 0.04 mm and a maximum of 0.57 ± 0.04 mm.

The SEM observations highlighted that when the sizes of the uncharged beads decreased, a localized wall subsidence could occur in the anchoring point during the support removal. An example of this is shown in Figure 5A–B. This phenomenon was not observed for the reinforced beads. In this latter case, Figure 5C and Figure 4D show the non-uniform distribution of HA, where the reinforcing particles were more numerous at the holes’ inner surfaces. The laser’s path and characteristics explain the higher concentrations of HA surrounding the holes. The laser can be characterized by a gaussian distribution of the energy across the spot area (TEM_00_ laser). Typically, the photopolymer resin requires a UV inhibitor, and its function is to prevent UV rays from penetrating excessively into the already-consolidated layers, which would cause solidification of the resin in undesired areas. If Equation (2) is considered, the presence of HA could partially diminish the energy available for the photopolymer, and therefore, at the periphery of the spot, where the incident energy was lower because of the Gaussian distribution, the reinforced resin could not reach an energy value sufficient to trigger polymerization. This would turn into a different degree of polymerization in the core and contour areas. Around the hole, the material under the peripheral spot area toward the center of the hole would not be completely polymerized, while that at the antipodes (but facing the fill zone) would undergo complete polymerization due to the adjacent passes in the core region. It can be conjectured that, in the washing phase, the non-polymerized material would be removed, whereas the HA may remain attached to the solidified resin. The post-processing phase in the UV oven would then consolidate the bond. Therefore, the observed higher concentration of HA around holes may have been due to the subtraction of the matrix from the boundary of the hole (Figure 6).

Figure 7 shows that the surface roughness of the beads’ upskin surface was between 2.38 and 4.76 μm. A slight trend of higher surface roughness for the reinforced formulations was observed, even when the geometry of the samples made the results uncertain. The surface maps in Figure 8 show a graphical representation of the bead topology.

The observation by the XmCT scan and the reconstruction of the bead model showed that although an internal alteration of the surface characteristics appeared in the zone where the staircase effect was most remarkable, the wall was substantially intact (Figure 9). The XmCT measurement highlighted the extreme regularity of the bead thicknesses, and the variation was in the order of a few hundredths of a millimeter.

The production rate depends on the bead size and post-treatment time. Vat photopolymerization allows for the manufacture of approximately 100–150 uncharged beads per hour, with a cost of approximately 0.3–0.4 euros per part. The production rate of charged beads decreases by approximately 15–20% due to the preliminary mix phases.

### 3.2. Beads Manufactured by PBF

Figure 10 compares the nominal diameter with the measured diameter using an optical microscope from different points of view, namely, along the Z-direction (build axis in the PBF machine) and along the X- and Y-direction. The dimensional deviation versus the nominal geometry did not show any specific anisotropy. The deviation of the nominal size from the measured one was influenced by the geometry. Types D and K had a deviation of −3% to + 1%, while the deviation of type V was + 1% to + 10%.

Table 6 shows the morphometric parameters obtained by the XµCT analysis of the beads with K-shapes (see Table 4). The results of the XmCT scans showed that the beads manufactured by PBF had two types of porosity: the first (open porosity) was the one desired and defined in the design phase and was due to the geometry of the bead, while the second was due to the process parameters, which caused incomplete densification. The latter (closed porosity) was the porosity inside the strut and may have been detrimental to mechanical performance. The open porosity values were compliant with the nominal ones and were within the experimental error. The open porosity results of higher than 70% for all the K-shaped beads analyzed was in agreement with the values favoring cell immobilization [31]. The percentage of closed porosity was below 0.5% for all the samples, which accounts for a high degree of densification (a density of > 99.5%). 

The diameters of the thicker struts were nearly half of the nominal value, while the thinner struts were produced with better accuracy. The closed porosity increased in accordance with the thickness of the strut.

The images reconstructed by XmCT showed that some struts were broken and others had non-constant sections and discontinuities closed to the nodes (Figure 11 and Figure 12). The defects at the nodes may have been due to the residual stresses induced by the various strut directions that, during shrinkage, act as ties. Figure 13 shows the XmCT-rebuilt models of the 30 mm K-shaped beads, where a large quantity of the unconsolidated powder was still present at the center of the beads, even after several dry-cleaning cycles using compressed air. It is worth noting that the peripheric regions, far from the scaffold core, showed open pores that were free of residual powder. In a previous contribution of the authors, Gatto et al. [32] found a similar behavior in PBF Ti6Al4V scaffolds built with different strut thicknesses, and they attributed the difficulty to emptying the structures toward the center to the geometry of the bead. From the data shown in Table 6, it was evident that the volume of the residual powder inside the structures was strongly dependent on the geometry, with the K30-1.0 beads containing approximately six times the quantity of residual powder of the K30-0.6 beads. The thicker struts may have acted as obstacles to the removal of the powder entrapped inside the beads. Furthermore, the difficulty in removing this residual powder may also be attributed to the partial sintering. The trabecular structure of the PBF bead may have caused a higher accumulation of heat during the building phase, in particular, in the core area. The heat caused the unconsolidated powder inside the bead to partially sinter, and thus, necks were generated between the various powder particles, and their removal became very difficult.

To summarize the feasibility of the beads made using PBF, the manufacturing limits are primarily attributable to the unconsolidated powder-emptying. The poor removal repeatability depended on the structure geometries and the partial sintering of the powder inside the beads. The widespread presence of defects, such as the strut thickening or the ruptures induced by residual stresses, depended on the heat distribution during the process. The tests showed that the production limit was not due to the minimum achievable definition, but rather, to the emptying capacities of the shapes produced.

The measured surface roughness of the beads on the upskin surface was between 5.66 and 9.06 μm.

The production capacity of the beads built by PBF ranged between 6 and 60 parts per hour, depending on the bead dimensions, with a cost of between 0.50 and 4 euros per part.

### 3.3. Discussion

Overall, the production of beads for cell immobilization by AM, as conjectured in the previous literature [13], proved to be viable using the two different material/process combinations. The results represent a remarkable advancement, as they open up the possibility of designing and producing complex geometries with diverse exchange surfaces, which could not be accomplished using the traditional methods [1,14,28,29]. As compared to the only previous study on beads produced by PBF [2], the present results show that the de-powdering can be a weak point, especially for certain bead shapes and strut thicknesses, due to undesired thermal accumulation in the bead core region. Vat photopolymerization is, instead, fully practicable, and it opens the possibility of creating complex shapes in a smaller size range that is considered optimal for the envisaged applications [13].

## 4. Conclusions

Vat photopolymerization proved to be a viable method for the production of beads with a minimum hole size of 200 µm and with a satisfactory surface roughness. By using resin reinforced with HA, a higher accuracy was obtained as the filler acted as a UV blocker, but the HA concentration was not uniform at the hole boundaries. With unreinforced resin, the bead accuracy declined, especially for the smallest bead sizes.

PBF, as a solution for the manufacture of beads, evidenced several limitations. The most critical aspect was the difficulty with unsintered powder emptying as a consequence of thermal accumulation and partial powder sintering for some bead geometries. The widespread presence of defects such as strut-thinning and ruptures was also observed. For PBF, the production capacity ranged between 6 and 60 parts per hour, depending on the bead dimensions, with a cost of between GBP 0.50 and 4 per part. Vat photopolymerization proved to have better feasibility, with a production capacity of approximately 100–150 parts per hour and a cost of approximately GBP 0.3–0.4 per part. In the scenario of an expected decrease in AM costs in the near future, the economic viability of the studied solutions may soon be achieved.

## Figures and Tables

**Figure 1 bioengineering-10-00150-f001:**
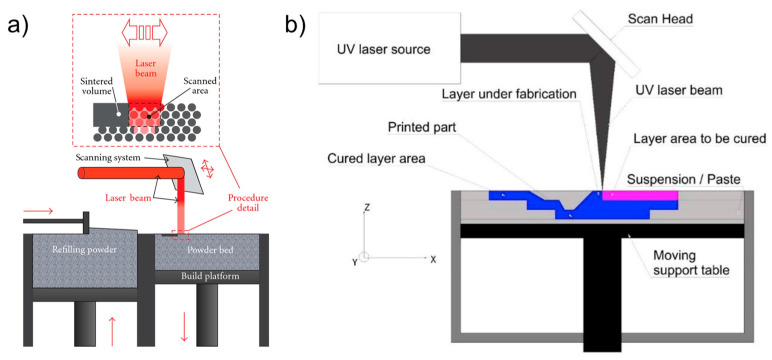
The systems used to build the specimens: (**a**) PBF [22] and (**b**) vat photopolymerization [23].

**Figure 2 bioengineering-10-00150-f002:**
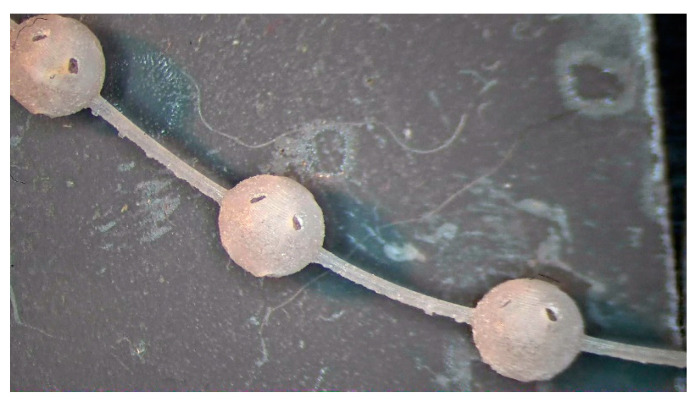
Vat photopolymerization: 3 mm beads built as a continuous chain arranged along a spiral on the build platform.

**Figure 3 bioengineering-10-00150-f003:**
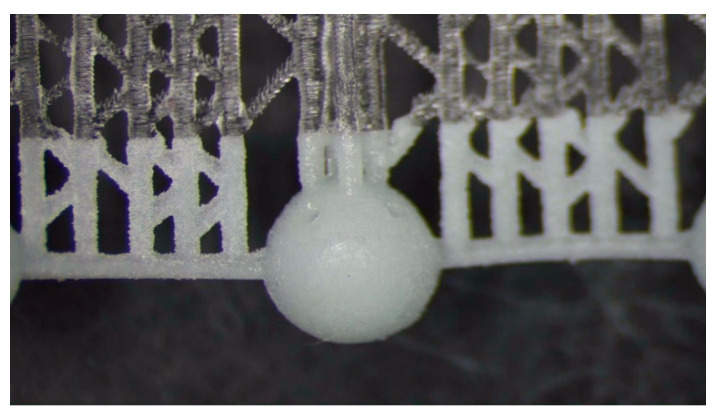
Vat photopolymerization: 3 mm beads built as a continuous chain and supported by a trabecular structure.

**Figure 4 bioengineering-10-00150-f004:**
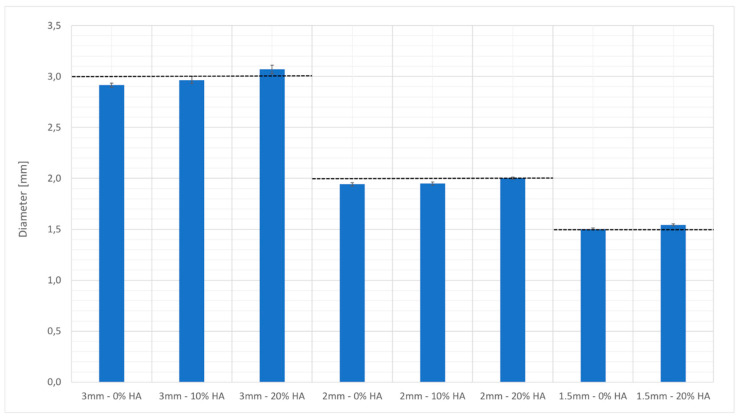
Nominal diameter versus the real diameters of the beads built by vat photopolymerization.

**Figure 5 bioengineering-10-00150-f005:**
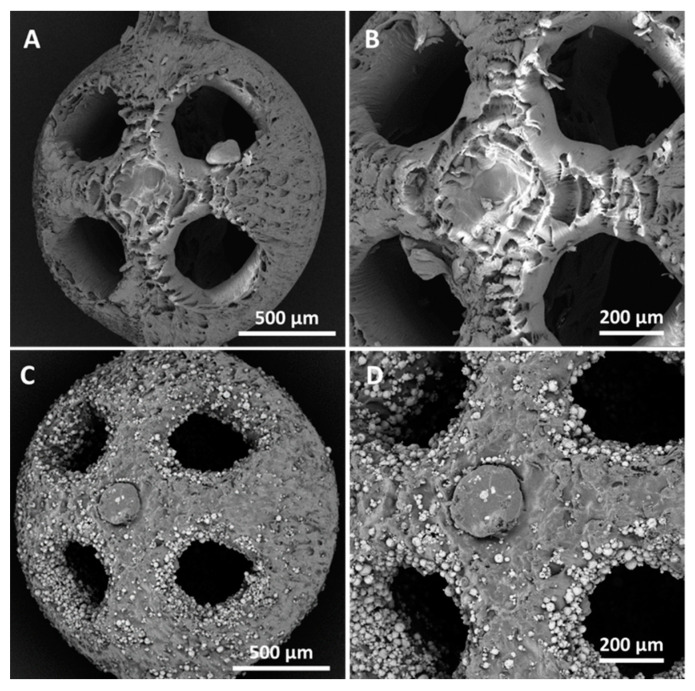
SEM images of the beads (d_bead_ = 2 mm and d_hole_ = 500 µm) manufactured by vat photopolymerization with acrylic acid ester, both unreinforced (**A**,**B**) and reinforced with 10% HA (**C**,**D**). The SEM micrographs in (**B**,**D**) show the details at higher levels of magnification.

**Figure 6 bioengineering-10-00150-f006:**
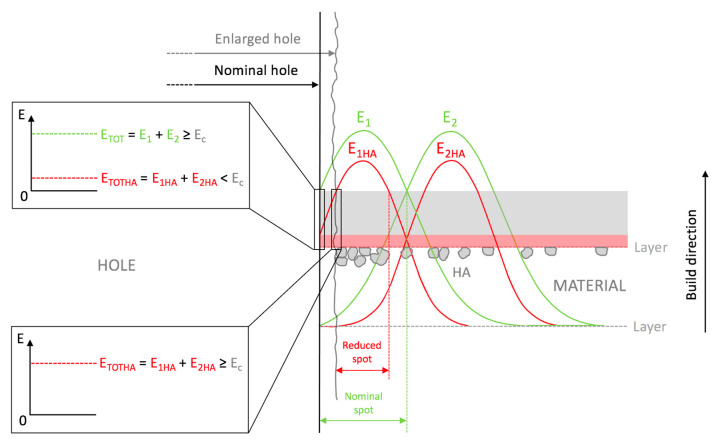
The presence of HA may partially diminish the absorbed energy; therefore, the material under the periphery of the spot may not have reached an energy value such as to trigger polymerization.

**Figure 7 bioengineering-10-00150-f007:**
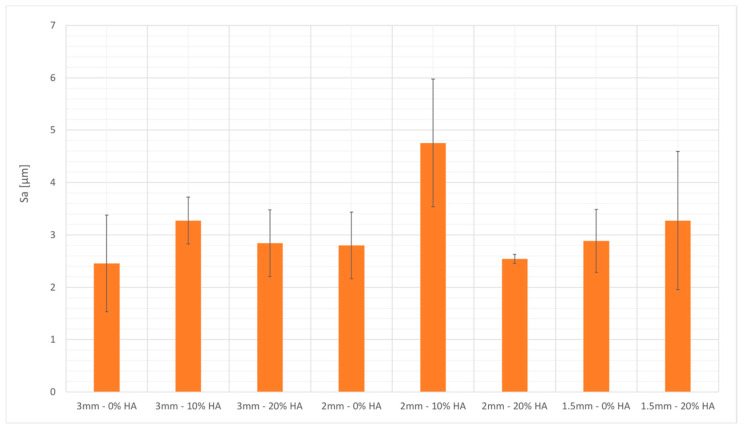
Surface roughness measurements of the beads built by vat photopolymerization.

**Figure 8 bioengineering-10-00150-f008:**
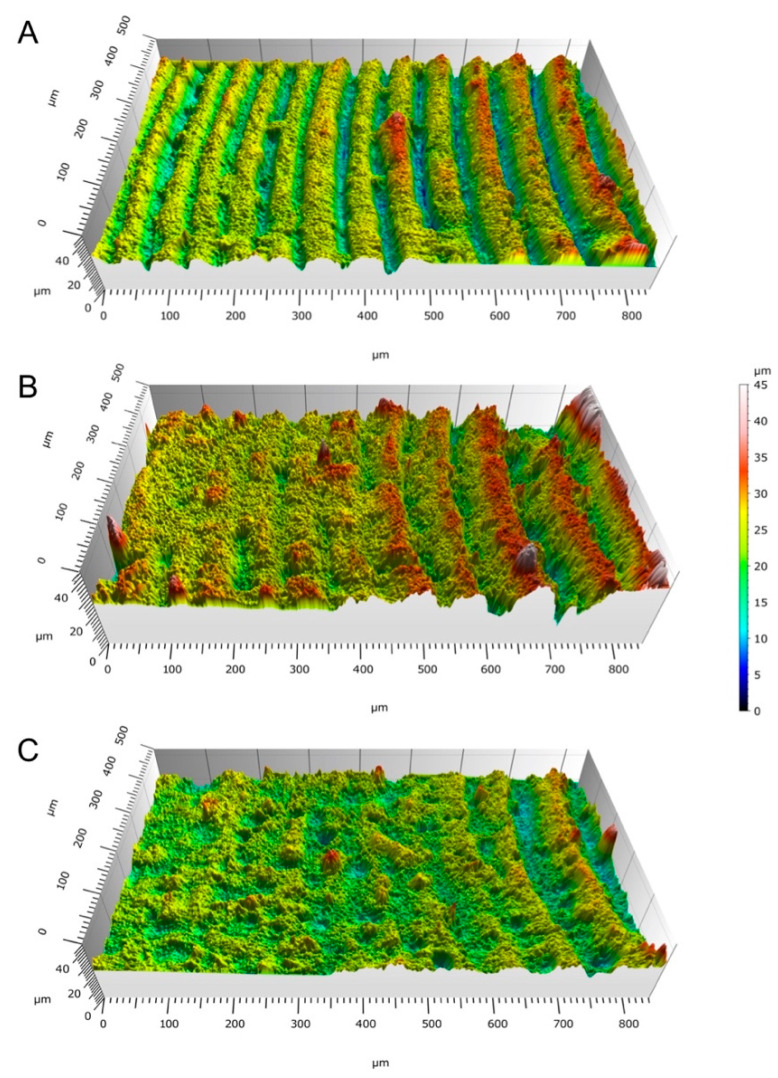
3D point cloud maps of the surfaces of the 3 mm vat photopolymerization beads with (**A**) 0% HA, (**B**) 10% HA, and (**C**) 20% HA.

**Figure 9 bioengineering-10-00150-f009:**
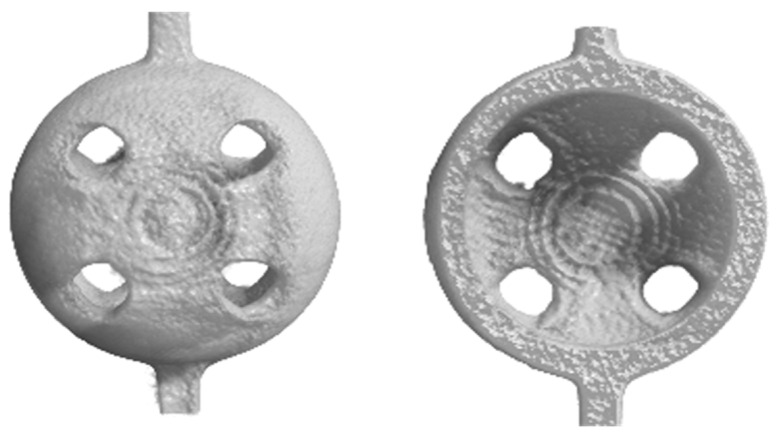
Acrylic acid ester beads (d_bead_ = 2 mm and d_hole_ =500 µm) manufactured by vat photopolymerization: XmCT scan and model reconstruction.

**Figure 10 bioengineering-10-00150-f010:**
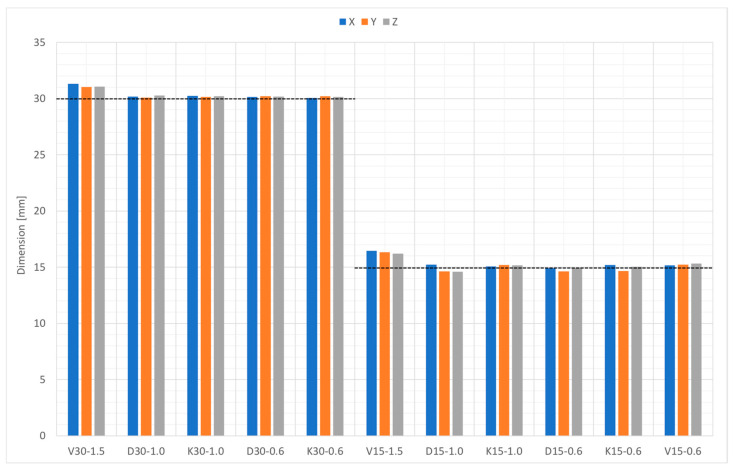
Bead dimensions in the X-, Y-, and Z-direction.

**Figure 11 bioengineering-10-00150-f011:**
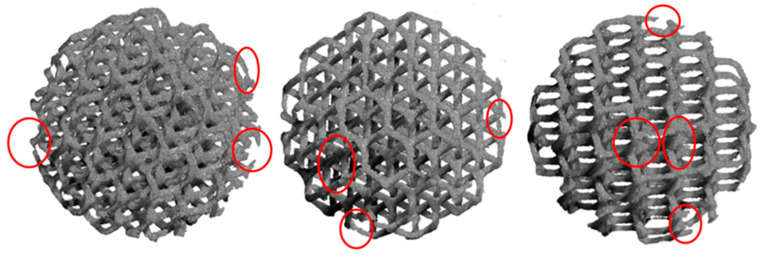
Models rebuilt by XmCT of the D15-0.6 beads. Broken rods, rods with uneven sections, and discontinuities at the nodes are visible.

**Figure 12 bioengineering-10-00150-f012:**
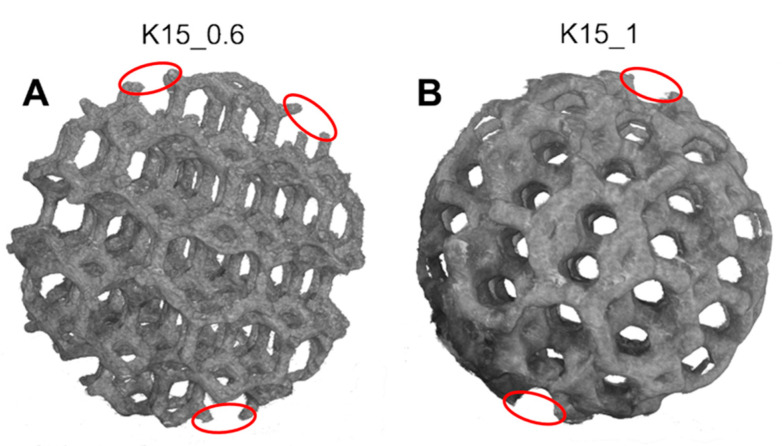
Models rebuilt by XmCT of the PBF K-type beads: (**A**) 0.6 mm strut diameter and (**B**) 1 mm strut diameter. Several defects are visible.

**Figure 13 bioengineering-10-00150-f013:**
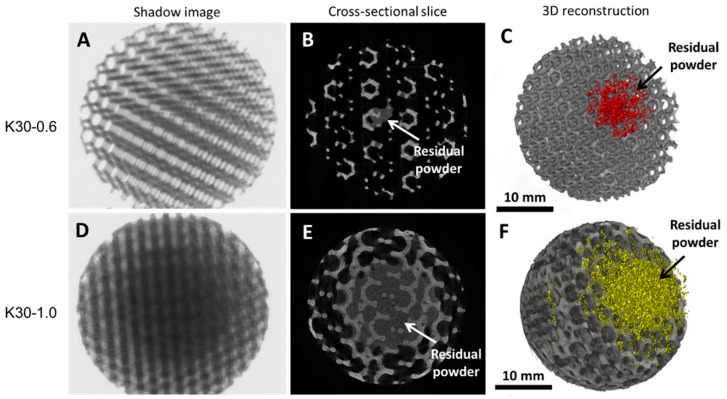
XmCT shadow images, 2D cross-sectional slices, and 3D reconstructions of the K30-0.6 beads (**A**–**C**) and the K30-1.0 beads (**D**–**F**), respectively. The arrows indicate the residual powder inside the bead cores.

**Table 1 bioengineering-10-00150-t001:** Nominal characteristics of the used materials.

		Polyamide 12 [24]	Acrylic Acid Ester [25]
Average particle size	(µm)	60	Not applicable
Apparent specific weight	(g/cm^3^)	0.435 ÷ 0.445	Not applicable
Sintered density	(g/cm^3^)	0.90 ÷ 0.95	Not applicable
Tensile modulus of elasticity	(MPa)	1700 ÷ 1500	1700 ÷ 2200
Tensile strength	(MPa)	45 ± 3	45 ÷ 55
Elongation at break	(%)	20 ± 5	6 ÷ 10
Flexural modulus of elasticity	(MPa)	1240 ± 130	2000 ÷ 2500
Resilience according to Charpy	(kJ/m^2^)	53 ± 4	-
Hardness	(Shore D)	75 ± 2	-
Fusion point	(°C)	172 ÷ 180	Not applicable
Viscosity at 25 °C (before photopolymerization)	(mPa ×s)	Not applicable	1000 ÷ 1400
Density (before photopolymerization)	(g/cm^3^)	Not applicable	1.01

**Table 2 bioengineering-10-00150-t002:** Geometrical characteristics of the beads produced by vat photopolymerization.

Bead Diameter(mm)	Shape	HoleDiameter(μm)	WallThickness(μm)	InternalVolume(mm^3^)	InternalSurface(mm^2^)	Exchange SurfaceContainment Surface(%)	Hydroxyapatite(Weight %)	SupportStrategy
3	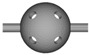	500	300	7.24	18.1	10	0	C ^1^
3	500	300	7.24	18.1	10	10	C ^1^
3	500	300	7.24	18.1	10	20	T ^2^
2	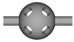	500	300	1.44	6.2	34	0	T ^2^
2	500	300	1.44	6.2	34	10	C ^1^
2	500	300	1.44	6.2	34	20	T ^2^
1.5	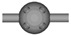	350	200	0.38	3.8	25	0	T ^2^
1.5	350	200	0.38	3.8	25	20	T ^2^

^1^ continuous spiral chain; ^2^ trabecular.

**Table 3 bioengineering-10-00150-t003:** Process parameters used for the beads produced by vat photopolymerization.

		Anchor Plate	Supports	Beads
Contours	(n)	3	3	3
Hatching	(mm)	0.5	0.5	0.5
Laser speed	(mm/min)	260	2800	5200
Layer thickness	(mm)	0.05	0.05	0.05
Number of layers	(n)	4	15	
Wavelength	(nm)	405		
Laser spot	(μm)	40		

**Table 4 bioengineering-10-00150-t004:** Geometrical characteristics of the PA 12 beads produced by PBF.

Sample ID	Shape	Sphere Diameter(mm)	Strut Thickness(mm)
D15-0.6	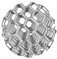	15	0.6
D15-1.0	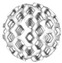	1.0
K15-0.6	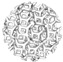	0.6
K15-1.0	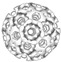	1.0
V15-0.6	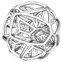	0.6
V15-1.5	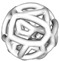	1.5
D30-0.6	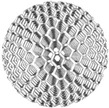	30	0.6
D30-1.0	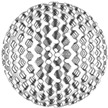	1.0
K30-0.6	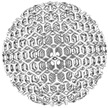	0.6
K30-1.0	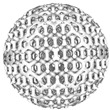	1.0
V30-1.5	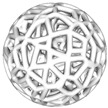	1.5

**Table 5 bioengineering-10-00150-t005:** Process parameters used for the beads produced by PBF.

System		EOS Formiga 110
Laser power	(W)	30
Laser type	-	CO2
Laser speed	(mm/s)	≈4500
Layer thickness	(μm)	100

**Table 6 bioengineering-10-00150-t006:** The beads’ morphometric parameters determined by XµCT analysis. The nominal values for strut thickness, volume, and open porosity are reported for reference.

Morphometric Parameters	K30-0.6		K30-1.0		K15-0.6		K15-1.0	
	AV	SD	AV	SD	AV	SD	AV	SD
Nominal strut thickness (mm)	0.6		1		0.6		1	
Nominal volume (cm^3^)	14.1		14.1		1.8		1.8	
Nominal open porosity (%)	88		71		89		71	
Average strut thickness (mm)	0.48	0.04	0.74	0.25	0.48	0.04	0.74	0.25
Average pore size (mm)	2.6	0.1	1.81	0.04	2.6	0.1	1.81	0.04
Closed porosity (%)	0.12	0.03	0.37	0.08	0.12	0.03	0.37	0.08
Open porosity (%)	87	1	70	1	87	1	70	1
Volume of residual powder (cm^3^)	0.2		1.2		-		-	

## Data Availability

The data are available upon request.

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
