# Peer review of "Beads for Cell Immobilization: Comparison of Alternative Additive Manufacturing Techniques"

_bioengineering, 2023, doi:10.3390/bioengineering10020150_

Round 1

Reviewer 1 Report

The manuscript explores an alternative additive manufacturing technique for cell immobilization in beads. Before it can be considered for publishing, some points must be improved:

·       Abstract needs improvement. It is not clear if it is a review or a research paper. As a research article, please add the main results and conclusion of the work

·       Please provide references for the sentence: “Calcium alginate is the most common immobilization matrix used today (also for wastewater treatment), followed by Polyvinyl alcohol (PVA).”

·       Authors studied 1.5-3mm and 15-30mm diameters. The diameter range is very different, how authors chose these conditions? Polyamide 12 and Acrylic acid esters have different properties, why not study them in the same sizes?

·       In general, methodology needs to be more detailed. For example, in line 120 how long the beads were UV oven?

·       Lines 128-132: please check the units in this paragraph

·       Table 2: please add the meaning of HA in the table footnote.

·       Please provide references for beads characterization methodology

·       Equation 2 should be in methodology, please move it.

·       In general, results are only presented in items 3.1 and 3.2. A comparison and discussion with literature is necessary. In addition, how is the comparison of the developed beads with data from literature of the traditional alginate ones?

·       Conclusions should be a concise paragraph with the main achievements of the work, not in topics. In addition, it is too long as it is. Please rewrite it in a continuous text way.

Reviewer 2 Report

The MS describes the production and characteristics of beads using additive Manufacturing for additions or entrapments of microbial cells and enzymes. It targets to support biotechnologists with an industrial perception. The MS consists of two manufacturing methods namely, beads manufactured by VAT photopolymerization and beads manufactured by PBF, and their full characterizations. The MS is written in a clear and lucid manner and provides satisfactory information about the manufacturing and characterizations. However, since the aim of bead manufacturing is for microbial cells and enzymes, it is expected to demonstrate the usefulness and compatibility with such materials of biological origins. That is because the conditions and photopolymerizations could have detrimental effects on biological materials, and therefore the whole manufacturing process may not be compatible with the biologics. Furthermore, the mass transfer ratio along with providing flexible environments or not restricting the conformational freedom of enzymes is a must for any entrapment matrix and directly proportional to the process efficiencies, an additional experiment to demonstrate or provide additional data could be very useful for completeness of the MS.   

Some other minor typos:

Line 131-132:…mm should be um: “in the design of bead, the hole dimension was increased to 500 mm for 3 and 2 mm 131 diameter beads, and to 350 mm for 1.5 mm

Reviewer 3 Report

This paper studies beads for cell immobilization, which has potential application value in engineering. In order to meet the requirements of high-quality publication of the journal, it is recommended to consider the following suggestions,

1) There is no quantitative data in Abstract Section.

2) The innovation of this article is not reflected in the first section and needs to be modified.

3) The second Section needs to add pictures of the experimental device.

4) What is the basis for selecting the parameter level in Tables 2~3?

5) The mehtod proposed in this paper needs to be compared with the previous literature, otherwise it cannot reflect innovation.

6) The Discussion Section needs a separate section.

7) The conclusion is too wordy and needs to be simplified.

8) There are few references in the last three years.

9) Does the format of References 15 and 16 meet the requirements of the journal?

Reviewer 4 Report

Generally, very interesting and potentialy important manuscript. Authors have done good job however there are still few thing that need to be corrected or added in order to improve manuscript quality. Comments are given in two document due to computer change. Second computer had Adobe reader without possibility of inserting comments. Thereforre first part of review is given as attachment in PDF, while the second part of review is given below:

Comments

Line 86: beads manufactured by VAT photopolymerization

Define VAT

Line 128: Preliminary tests were performed to verify the feasibility of 200 mm holes

Does the authors mean 200 μm holes?

Lines 131-132: in the design of bead, the hole dimension was increased to 500 mm for 3 and 2 mm 131

diameter beads, and to 350 mm for 1.5 mm beads.

Does the authors mean 500 μm holes and 350 μm holes

Table Sphere diameter Strut thickness

give dimensions for those parameters [mm]

Explain why surface roughness is important factor for cell immobilization.

Are there some examples of cell immobilizations in types of beads exanimated within this work? Are there some examples that they are applicable for cell immobilization? If there are examples, introduce them, if there is not stress this in the introduction.

Although authors state in the abstract that those methods are good for industrial scale production price and production rate are still not so favorable and need to be further improved and this need to be stressed.

Round 2

Reviewer 1 Report

The authors have responded all the raised points accordingly. The manuscript can be published.

Reviewer 2 Report

The MS still needs improvements addressing the microbial cells, enzyme issues, and compatibility with them.

Reviewer 3 Report

The authors have addressed all my concerns.